# Bayesian spatial modelling of localised SARS-CoV-2 transmission through mobility networks across England

**Thomas Ward**[1]*, **Mitzi Morris**[2], **Andrew Gelman**[3], **Bob Carpenter**[4], **William Ferguson**[1], **Christopher Overton**[1,5], **Martyn Fyles**[1]

**1** UK Health Security Agency, Infectious Disease Modelling Team, London, United Kingdom, **2** The University of Columbia, Institute for Social and Economic Research and Policy, New York, New York, United States of America, **3** The University of Columbia, Department of Statistics, New York, New York, United States of America, **4** The Flatiron Institute, Centre for Computational Mathematics, New York, New York, United Kingdom, **5** The University of Liverpool, Department of Mathematics, Liverpool, United Kingdom

* Tom.Ward@UKHSA.gov.uk

**Data Availability Statement:** The data used in this study is not publicly available. UKHSA operates a robust governance process for applying to access protected data. Access to protected data is always

## Abstract

In the early phases of growth, resurgent epidemic waves of SARS-CoV-2 incidence have been characterised by localised outbreaks. Therefore, understanding the geographic dispersion of emerging variants at the start of an outbreak is key for situational public health awareness. Using telecoms data, we derived mobility networks describing the movement patterns between local authorities in England, which we have used to inform the spatial structure of a Bayesian BYM2 model. Surge testing interventions can result in spatio-temporal sampling bias, and we account for this by extending the BYM2 model to include a random effect for each timepoint in a given area. Simulated-scenario modelling and real-world analyses of each variant that became dominant in England were conducted using our BYM2 model at local authority level in England. Simulated datasets were created using a stochastic metapopulation model, with the transmission rates between different areas parameterised using telecoms mobility data. Different scenarios were constructed to reproduce real-world spatial dispersion patterns that could prove challenging to inference, and we used these scenarios to understand the performance characteristics of the BYM2 model. The model performed better than unadjusted test positivity in all the simulation-scenarios, and in particular when sample sizes were small, or data was missing for geographical areas. Through the analyses of emerging variant transmission across England, we found a reduction in the early growth phase geographic clustering of later dominant variants as England became more interconnected from early 2022 and public health interventions were reduced. We have also shown the recent increased geographic spread and dominance of variants with similar mutations in the receptor binding domain, which may be indicative of convergent evolution of SARS-CoV-2 variants.

strictly controlled using legally binding data-sharing contracts. UKHSA welcomes data applications from organisations looking to use protected data for public health purposes. To request an application pack or discuss a request for UKHSA data you would like to submit, contact DataAccess@ukhsa.gov.uk. The Stan model code can be found in the S1 Data file.

**Funding:** The authors received no specific funding for this work.

**Competing interests:** The authors have declared that no competing interests exist.

## Author summary

Emerging variants of SARS-CoV-2 have been a significant catalyst of epidemic waves of incidence globally. These variants have caused concern due to transmission advantages, changes in the infection severity profile and immunological evasion. Understanding the spatial dispersion of a novel variant can be obfuscated by limited geographic test coverage, the ascertainment rate, and the proportion of tests that undergo sequencing or genotyping. Therefore, we have developed a spatial modelling approach based on the changing mobility patterns across England. We have used a BYM2 model structure that has been extended to include a random effect on each timepoint to account for variable daily testing patterns. This approach was assessed using simulated epidemic scenarios in England that would accurately reflect the spatial patterns of an emerging variant. The model estimated variant prevalence more accurately than unadjusted test positivity in every scenario tested particularly, when testing coverage was low and missing for some locations. We have used this modelling approach to estimate the spatial dispersion and growth for every dominant variant since the start of the SARS-CoV-2 pandemic.

## Introduction

The COVID-19 pandemic has precipitated unprecedented public health interventions to reduce the transmission of the virus. Outbreaks of emerging variants have driven resurgent epidemic waves of incidence globally from SARS-CoV-2. The public health response to these outbreaks can be impacted by the limited understanding of emerging variants' spatial dispersion. This is a consequence of limited data at the start of an outbreak and geographical bias in testing coverage and reverse-transcription polymerase chain reaction (RT-PCR) sequencing or genotyping.

Localised geographic monitoring of SARS-CoV-2 transmission across the pandemic in the UK has been largely conducted through the widespread use of RT-PCR and lateral flow tests (LFTs) [1,2]. RT-PCR testing became available for all symptomatic individuals in the UK from May 2020 [3] and more accessible across the country as laboratory infrastructure was developed [4]. Geographic and socio-economic disparities in test availability were apparent at the start of the pandemic [5,6] and this became less significant with the Home Test Service (HTS) that delivered RT-PCR tests through the postal service. In October 2020, LFTs became available in a limited capacity, and began to be used more widely in April 2021 within institutional settings and for asymptomatic infections [7]. Public health policy initially required that positive LFTs were confirmed with an RT-PCR test although this policy was later dropped. Considerable ascertainment bias was an inevitable consequence of this test by request strategy, which impacted the representation of minority and lower socio-economic groups [5] in the testing data. Targeted or surge testing was conducted in localised areas with high levels of transmission throughout the pandemic [8] or where novel variants had been detected [9]. This hampered the spatial and temporal understanding of localised prevalence as some areas were disproportionately sampled. Free mass testing came to an end in April 2022 [10] and consequently RT-PCR tests are only conducted for the clinically extremely vulnerable, in hospital settings, and until the end of March 2023 through the Office of National Statistics Community Infection Survey [11].

Resurgent outbreaks of COVID-19 have been largely driven by more competitive variants of SARS-CoV-2 and changes in the policy of non-pharmaceutical interventions (NPIs). The spatial patterns of emerging variant introductions have been heterogeneous and influenced by

the site of emergence and the number of seeding events. The Alpha variant, which was first detected in England from a sample in September 2020, caused the second wave of incidence of SARS-CoV-2 infections in the UK [12]. Transmission of this variant initially clustered in the county of Kent and then spread throughout England but was particularly concentrated in the South East of England [13]. The Delta variant began replacing Alpha [14] in April 2021, which was during a time of national NPI easing. Delta was largely introduced through international travel from India, where it was first detected in late 2020 [15]. The strict NPIs that were enforced, slowed the rate that Delta replaced Alpha, leading to one of the longest periods of dominant variant replacement [16]. Despite the long period of time it took for Delta to become dominant, early cases of this variant were spatially dispersed due to many early seeding events across England [14]. The Delta period was characterised by oscillations in incidence, which was likely a result of NPI easing throughout 2021.

Omicron BA.1 was first detected in South Africa in November 2021 and importations of this variant were detected in England by the end of the month [17]. The outbreak of Omicron BA.1 was initially concentrated in the South East of England, with a considerable proportion of the international importations into this area. It had the shortest replacement period of any variant to date and had the greatest absolute growth rate [16]. Omicron BA.2, which was introduced in December 2021, began to replace Omicron BA.1 in January 2022 [17]. Then in May 2022, Omicron BA.4 and BA.5 began to grow in the UK, with an early growth rate advantage noted for Omicron BA.5 [18]. Despite Omicron BA.4 becoming dominant in South Africa in April/May 2021, it was largely Omicron BA.5 and its sublineages that went on to become dominant across the regions of England [19].

Understanding the localised spatial growth of emerging variants is obfuscated by disparities in geographic test coverage, the ascertainment rate of infections between areas, and the proportion of diagnostic tests that undergo sequencing or genotyping. Approaches to understand localised growth have included the use of generalised additive models to calculate the instantaneous growth rate for each Lower Tier Local Authority (LTLA) [14]. These models describe growth within LTLAs but do not account for spatial dispersion and can be limited by the ascertainment rate of tests being an accurate representation in the trend of incidence. Exceedance models [20] were commonly used by Public Health England to differentiate LTLAs based on the RT-PCR positivity rates. A limitation of this approach is that a meaningful baseline, which is required to calculate the exceedance, can be difficult to define particularly with unbalanced testing coverage across different geographic levels. The use of presence-absence modelling to understand the spatial spread of SARS-CoV-2 were explored by Smallman-Raynor et. al [21] and this approach can be useful to monitor the rate of spatial dispersion. However, this approach will not allow for the determination of within LTLA spatial growth or the differentiation in variant prevalence between locations.

In epidemiological modelling, mobility data plays a key role in performing large-scale spatio-temporal modelling of epidemics. Often, mobility data is used to construct spatial metapopulation models of epidemics, which use simulations to model the course of the epidemic, such as projections that estimate the impact of lifting lockdowns [22], investigating the effects of travel bans [23], and the spatial spreading of epidemics [24]. In this paper we seek to infer the spatial spread of emerging variants. Simulation-based approaches are not suitable for this problem given that parameter estimates for emerging variants are likely to be highly uncertain. Therefore, rather than using mobility data to perform simulations, we are interested in using mobility data to conduct inference and to inform prior distributions. For example, Lemey et. al. used mobility data in conjunction with genetic sequence data to reconstruct the phylogenetic tree for SARS-CoV-2 [25] and influenza H3N2 [26]. Models such as the BYM and BYM2 are often used to conduct the mapping of a disease or inferring the spatial pattern of a feature

of a disease [27–29]. However, in such models it is common for the spatial correlation structure to be determined by either geographical distance, proximity, or adjacency. Given the geography of the UK however, and our goal of inferring the spread of an emerging variant, it is likely that these proxy choices are not appropriate for determining spatial relationships.

In this paper, using telecoms data, we derive mobility networks that describe the movement patterns between LTLA's in England, which we use to construct the spatial structure of a Bayesian BYM2 spatial model. We have performed spatial modelling of emerging variant growth using whole-genome sequencing, and genotyped RT-PCR tests. We further assess the performance of the model through simulated epidemics with scenario analysis.

## Methods

All data were anonymised prior to data access and conducted in line with the UK government's response to COVID-19.

### Data

The data used in this study was aggregated to LTLA level geography, which is a local government geographic area defined by the UK government and there were 309 areas designated in England during the majority of this study. RT-PCR test results were sourced at LTLA from the Second Generation Surveillance System. RT-PCR whole-genome sequencing and genotyped tests were categorised through a rules-based decision algorithm for the confirmed and probable categorisation of variant and mutation (VAM) profiles from genotype assay mutation profiles. This information is provided at an individual level through the UK Health Security Agency (UKHSA) internal VAM linelist. Analysis of variants with receptor binding domain mutations which converge on the same site including: R346T, N460K, K444T, G446S, F486S, R346I, K444M, N450D, V445A, K444R, F490S, and F486P was also conducted. Population level data for LTLAs in England was provided by the 2011 population adjusted Office of National Statistics census [30]. Mobility data was sourced from a mobile telecom provider, which supplied the average number of journeys for each given hour in a day between origin and destination LTLAs by commuters and 'other' [31]. This data was aggregated to provide a single relative weighting for each origin-destination pair by week, covering the period of the study, which was the 12th November 2020 to the 27th October 2022. Maps were created from geographical files using the House of Commons Library, which is under the Open Parliament License v3.0 [32].

### BYM2 spatial model

We are interested in modelling the spread of emerging variants across England, given that we have various indicators of variant proportions with heterogenous coverage. Therefore, our aim is to use a modified BYM2 model [33,34] for spatially correlated count data with random effects to estimate the proportion of infections in an area that are due to a specific variant.

Each of the $n \in \mathbb{N}$ areas in our dataset are assigned an integer identifier, such that $A = \{1,..., n\}$ denotes the set of all areas. For each area we observe $T \in \mathbb{N}$ days of data. Hence, for a given area $i \in A$, and a given time point $t \in \{1,2,..,T\}$ we observe the pair $(N_{i,t}, y_{i,t})$ where $N_{i,t} \in \mathbb{N} \cup \{0\}$ is the number of positive tests and $y_{i,t} \in \{0,1,..,N_{i,t}\}$ indicate the number of test results that were infected with our target variant. Therefore, our aim is to model $p_{i,t} \in [0,1]$ using the likelihood function

$$y_{i,t} \sim \text{Binomial}(N_{i,t}, p_{i,t}),$$

where $p_{i,t}$ is the proportion of cases that have variant status in an area $i$ at time $t$. To model $p$ a generalised linear model with spatial correlation and random effects is employed, where $p = \text{logit}^{-1}(\alpha+\gamma)$. As in standard generalised linear models, $\alpha$ is our intercept term, whereas the $\gamma$ vector is the contribution from a combined spatial and random effects model that we adapt to our setting from the work of Riebler et. al [33] who developed the BYM2 model. The contribution of the BYM2 model over the original BYM model was to re-parameterise the combined spatial and random effects such that it is easier to set priors on the relative contribution of the spatial effects versus the random effects.

The formulation of BYM2 does not include a time dimension, and as such the combined spatial and random effect for an area $i$ is defined using

$$\gamma_i = \theta_i + \phi_i = \sigma\left(\sqrt{\frac{r}{\tau}} \cdot \theta_i^* + \sqrt{1 - r} \cdot \phi_i^*\right),$$

where $\theta \in \mathbb{R}^n$ is the spatial component, and $\phi \in \mathbb{R}^n$ is the random effects component. These components are a rescaling of $\theta^*, \phi^* \in \mathbb{R}^n$ which are the spatial and random effects components respectively scaled to have unit variance. As such, the overall variance of the combined spatial and random effects is controlled by $\sigma \in \mathbb{R}_+$, and $r \in [0,1]$ controls the relative contribution of spatial effects against the random effects. The parameter $\tau$ is a scaling component adjusting for the fact that the random effects and the spatial effects occur on different scales and allows for meaningful priors to be set.

For our data, a key concern is the presence of surge testing effects that can cause sampling bias. It is possible many tests could be conducted over a short period of time and in specific locations that may be at high risk of being infected with a single variant. One example might be that a variant of interest is detected in a care home, which could lead to the testing of all members of that care home resulting in many tests that are positive for that variant of interest. Given the scale of surge testing in comparison to regular testing, it is possible for a small number of days of data to skew the sample prevalence for a given area, therefore we consider it appropriate to model the data at a daily resolution, and to adjust for random effects.

Consequently, we find it necessary to extend the BYM2 model to include a daily-level random effect for each LTLA via

$$\gamma_{i,t} = \theta_i + \phi_i^{(1)} + \phi_{i,t}^{(2)} = \sigma\left(\sqrt{\frac{\rho_1}{\tau}} \cdot \theta_i^* + \sqrt{\rho_2} \cdot \phi_i^{(1),*} + \sqrt{\rho_3}\phi_{i,t}^{(2),*}\right),$$

where $\theta$ is an LTLA level spatial component, $\phi^{(1)}$ is an LTLA-level random effect component and $\phi^{(2)}$ is an LTLA-time level random effect component. The three different components are a rescaling of $\theta_i^*, \phi_i^{(1),*}, \phi_{i,t}^{(2),*}$ which are constrained to have unit variance. The proportion of variance attributed to each component is controlled via $\rho \in \mathbb{R}_+^3$ which is constrained such that $\sum_{i=1}^3 \rho = 1$ (i.e., a simplex). This parameterisation maintains the properties that Riebler et. al. [33] identified as being advantageous, while incorporating area-time level random effects.

For the spatial component $\theta^*$ an intrinsic conditional autoregressive (ICAR) random variable is used—a degenerate case of conditional autoregressive (CAR) random variables that are commonly used in spatial modelling. The distribution of the elements of $\theta^*$ are defined conditional upon a weighted sum of the other elements of the random variable:

$$\theta_i | \theta_{-i} \sim \mathcal{N}(\sum_{j \neq i} w_{ij}\theta_j, v),$$

where $w_{ij} \in \mathbb{R}_+$ represents the spatial correlation between two areas, and $\theta_{-i}$ is $\theta$ with the $i^{th}$ element omitted. This conditional definition of the ICAR random variables results in a Gaussian

**Table 1. Prior distributions for key model parameters.**

| Parameter | Prior Distribution |
|---|---|
| α | $\mathcal{N}(0, 1)$ |
| β | $\mathcal{N}(0, 1)$ |
| σ | $\mathcal{N}(0, 1)$ |
| r | Dirichlet(2,2,2) |
| $\phi^{(1),*}$, $\phi^{(2),*}$ | $\mathcal{N}(0, 1)$ |
| θ* | ICAR prior with unit variance and constrained to sum to 0 |

Markov random field, defined on a weighted graph structure that we need to construct. The spatial component is further constrained to sum to zero to ensure identifiability of the model. The prior distributions for the model parameters can be seen in *Table 1*.

As an input to the ICAR random variables we construct a graph of our LTLAs, where each LTLA is represented as a vertex, and two LTLAs are connected by an edge if we consider them to be connected in some sense. If two regions are not connected by an edge, then this implies $w_{ij} = 0$ in the ICAR random variable. This graph encodes the spatial correlation structure of our data in an adjacency matrix which we use as an input for computing the distribution of our ICAR random variables. The spatial spread of human-to-human transmitted pathogens is, by its nature, facilitated by individuals from one area visiting another area and infecting or being infected. This suggests that methods for constructing the edge network that do not account for mobility patterns will not truly capture the spatial correlation of the pathogen. For example, one common method of constructing the edge network is spatial adjacency, where two areas are connected if they are adjacent to one another, however this method could in theory connect two areas where there is little to no human mobility between those areas. Alternatively, commuter towns where a substantial portion of the population commute to another area to work may not be directly connected if there is no spatial adjacency, despite these commutes coupling the epidemic in the two areas. As such, we have obtained and used telecoms mobility data to construct a graph of how individuals move between areas, termed the mobility rate graph. The resulting graph is weighted, and undirected, with no self-edges allowed.

We introduce sparsity in the mobility rate graph to keep computational times reasonable, by removing edges if $w_{ij} < 0.001 * max(W)$ where W is the matrix of unweighted edges prior to thresholding. The resulting graph consists of a single component, with an edge density of 15.6%, and each area is connected to 48 other areas on average. In *Fig 1* we visualize the sparse edge network, where edges with low weights have been removed, overlaid on a hex map of England. The size of each LTLA in the hex map corresponds to size of the population in that area. Each edge is plotted between the centroids of different LTLA's, with the intensity of the edge weight corresponding to the rate of journeys between those two LTLA's, normalised to take values in [0,1]. For comparison, the spatial adjacency-based network graph has an edge density of 1.6%, and each area is connected to 5.2 other areas on average. As discussed previously, the scaling factor τ must be computed for this, and we find that $\tau_{mobility} = 1.981$.

The model is implemented in Stan [35], and we use Hamiltonian Monte Carlo to draw samples from the posterior. We use 4 chains, discarding the first 2000 samples from each chain as warm up, and each chain draws 1000 samples from the posterior. Model convergence is assessed through potential scale reduction factor or $\widehat{R}$ where a value less than 1.01 is desirable.

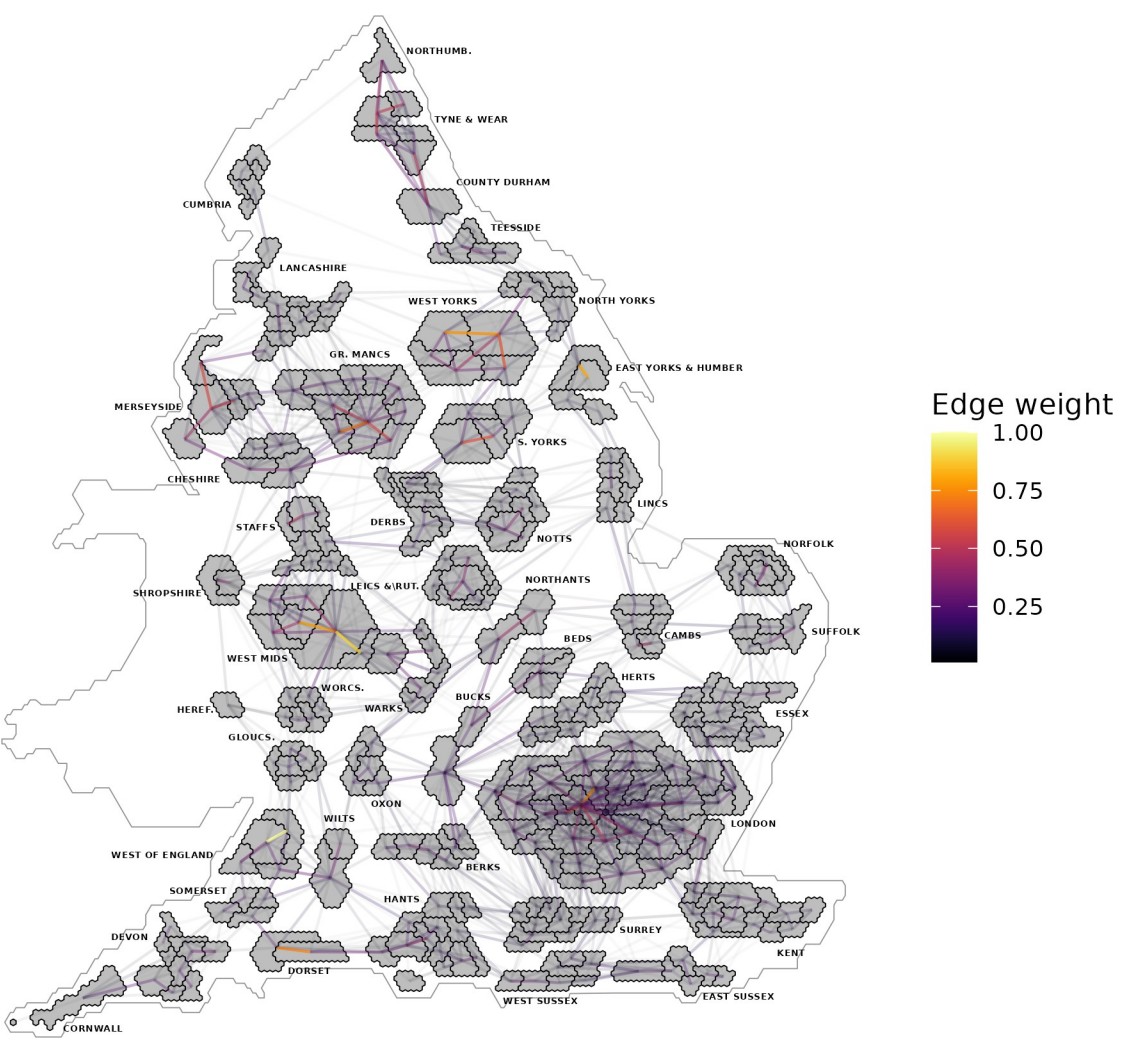

**Fig 1. The mobility network overlaid on a hex map of England, where the edge weight describes the rate of journeys between different LTLA's, normalised to take values in [0,1].** Created from geographical files using the House of Commons Library, which is under the Open Parliament License v3.0.

## Analysing the spread of variants across England

The BYM2 model can be fit for various time periods, typically for an infectious disease a meaningful window is in the order of a week and up to a month. We modelled SARS-CoV-2 variants using the proportions of sequenced or genotyped RT-PCR tests, with variant designations, across defined temporal windows.

## Synthetic metapopulation analysis

A key goal of our work is to understand the robustness and performance of the inferences made by the BYM2 model, particularly when provided with challenging data scenarios that may arise as the SARS-CoV-2 response continues to evolve. We have identified the following as potential scenarios that may prove challenging for the BYM2 model and could potentially degrade its performance: extremely sparse data availability; large random effects due to targeted testing; and highly localised spread of variants, such as clusters of a new variant. To

explore model performance under these scenarios, we simulated the spatial spread of a synthetic SARS-CoV-2-like epidemic using a susceptible infected recovered (SIR) metapopulation model [36]. The model is at LTLA-level and spread between different LTLAs was parameterised using the mobility dataset.

Each individual in the metapopulation model is in one of three states: susceptible, infected, or recovered. We let $K = \{1,\dots,n\}$ represent a set of labels for each of the $n$ different areas in the model. Let the state of the epidemic in the $k^{th}$ area, $k \in K$ at time $t \in \mathbb{R}_+$ be is given by $A_k(t)$ which describes the size of the susceptible, infected and recovered subpopulations respectively, $A_k(t) = (S_k(t), I_k(t), R_k(t))$. Each area has fixed population given by $N_k = S_k + I_k + R_k$, and we do not model birth/death processes, or migration between areas.

The infection is propagated by a contact occurring between different individuals. Specifically, if there is a contact between an infected individual, and a susceptible individual, then there is a probability of transmission occurring which can result in the susceptible individual moving to the infected state. For within-area contacts, we assume homogenous mixing. This assumption implies that each case makes contacts at the same rate, and the recipient of those contacts are selected uniformly at random. For between-area contacts, we specify the total rates of contacts between those two areas using our mobility data, however we still assume homogenous mixing in the sense that given a contact has occurred, the two individuals making the contact are selected uniformly at random from each area.

Let $\beta_1 \in \mathbb{R}_+$ be the within-area force-of-transmission parameter, defined as the rate that a given infected makes infectious contacts–contacts that would lead to infection if the contact recipient is a susceptible individual. As the contacted individual is selected uniformly at random, for the $k^{th}$ area the probability that the infectious contact is to a susceptible individual is given by $\frac{S_k}{N_k}$. Therefore, since there are $I_k$ infected in the area, the rate at which susceptible individuals in area $A_k$ are infected due to within-area transmission is given by:

$$\beta_1 \frac{I_k S_k}{N_k}, \qquad k \in K$$

Next, we consider the rate at which susceptible individuals in area $A_k$ are infected due to contacts with infected individuals in area $A_j$, for $j, k \in K$. So that our model is parameterised using mobility networks, we assume that the rate at which contacts occur between areas $j, k$ is proportional to $w_{j,k}$. The probability that a contact is between an infected individual in area $j$, and a susceptible individual in area $k$ is given by $\frac{I_j S_k}{N_j N_k}$. If we assume that each contact causes infection with probability $\beta_2$, then the rate at which susceptible individuals in area $k$ get infected due to transmission from infected individuals in area $j$ is proportional to

$$\beta_2 w_{jk} \frac{I_j S_k}{N_j N_k}, \qquad j, k \in K, j \neq k$$

In practice, since we have assumed that $w_{jk}$ is proportional to the number of contacts, we include this proportionality constant in $\beta_2$, so that $\beta_2 \in \mathbb{R}_+$ rather than specifically being a probability. As a result, the above equation gives the exact rate at which susceptible individuals in area $k$ are infected due to contacts with infected individuals in area $k$.

When an infected individual recovers, they move from the infected state to the recovered state. For all infected individuals, recovery happens at constant rate $\gamma \in \mathbb{R}_+$. Once an infected individual is in the recovered state, they remain there and cannot return to the susceptible or infected state.

Putting this all together, we arrive at a system of ordinary differential equations that describe the spread of the infection. For $k \in K$ we have the state of the epidemic evolves according to

$$\frac{dS_k}{dt} = -\beta_1 \frac{S_k I_k}{N_k} - \beta_2 \cdot \sum_{j \in N(k)} w_{jk} \frac{I_j S_k}{N_j N_k},$$

$$\frac{dI_k}{dt} = -\gamma I_k + \beta_1 \frac{S_k I_k}{N_k} + \beta_2 \cdot \sum_{j \in N(k)} w_{jk} \frac{I_j S_k}{N_j N_k},$$

$$\frac{dR_k}{dt} = \gamma I_k,$$

where $N(k)$ denotes the neighbour set of area $A_k$, defined as areas that are connected to $A_k$ by an edge.

Two epidemics, one labelled 'variant A' and the second labelled 'variant B', are simulated using this metapopulation model. These are independent epidemics, and we do not simulate two variants competing for dominance. The spatial pattern of each synthetic variant is obtained by sampling the epidemics at different points in time. Testing data for each LTLA was generated by assigning probabilities that, during our assumed time period of interest, infected cases are tested with probability $\rho \in [0,1]$. We make the simplifying assumption that all tests of infected individuals return a positive, and that all tests of susceptible or recovered individuals return a negative. For an area $A_k$, let the number infected with variant A be given by $I_k^A$, and the number infected with variant B be given by $I_k^B$. The number of tests positive for variant A in area $k$ is given by $P_k^A \sim \text{Binomial}(I_k^A, \rho)$, and the number of tests positive for variant B in an area $k$ is given by $P_k^B \sim \text{Binomial}(I_k^B, \rho)$.

In order to add random effects into the data generating process, each area is assigned probabilities such that the $k^{\text{th}}$ area has probability $\tilde{\rho}_k^A$ of testing an individual who is positive with variant A, and probability $\tilde{\rho}_k^B$ of testing an individual who is positive for variant B. Both $\tilde{\rho}_k^A$ and $\tilde{\rho}_k^B$ are obtained by perturbing the baseline probability of testing a infected individual, $\rho$, from it's original value via $\tilde{\rho}_k^A = \text{logit}^{-1}(\text{logit}(\rho) + \xi_k^A)$ and likewise for variant B where $\xi^A, \xi^B \sim N(0, \sigma)$ for some value of $\sigma \in \mathbb{R}_+$.

The true proportion of variant B in area $A_k$ is given by $\eta_k := \frac{I_k^B}{I_k^A + I_k^B}$, and let $\widehat{\eta}_k$ be an estimate of $\eta_k$. We evaluate the performance of this estimate using the $L_1$ error defined as: $S_1 = \sum_{k=1}^n |\widehat{\eta}_k - \eta_k|/n$, which is interpreted as the average absolute difference between the true value, and the estimate across all areas. For the BYM2, we use the posterior mean as our estimate of $\eta_k$, which we denote as $\widehat{\eta}_k^{\text{bym2}}$. We compare the performance of the BYM2 estimate against a simple binomial proportion estimator given by $\widehat{\eta}_k^{\text{naive}} := \frac{P_k^B}{P_k^A + P_k^B}$, which we term the naïve estimator, given that it does not include any spatial autocorrelation.

The BYM2 model performance is then assessed under three main scenarios that describe different spatial patterns of variants, and as such present different challenges to conducting inference. For scenario 1, variant B has a relatively homogenous spread across England, and this scenario may simulate a variant that is unable to completely replace the existing dominant variant or is slowly replacing with little to no geographical heterogeneity. Scenario 2 introduces a more complicated spatial spread pattern, with variant B primarily concentrated in London, though with significant case numbers observed in most areas of England. This spatial pattern

is typical for variants that are being imported at high rates, as cases in London are expected to take off first due to its intrinsic connectedness and large population sizes, before reaching less well-connected areas of England. Scenario 3 simulates clusters of emerging cases that are highly localised to specific regions of England, as might be expected for a newly evolved variant undergoing rapid growth but has not yet disseminated across England. As such, scenario 3 represents the most challenging spatial spread pattern under which inference must be conducted. Moreover, estimating the localised spread could be further complicated if sequencing or targeting coverage is particularly poor in an area with a localised cluster.

The ability of the model to conduct inference under each scenario is first conducted under ideal conditions, and then stress-tested by exploring model performance under modified versions of the scenario that have small sample sizes resulting in several areas with missing data, large random effects, and simultaneously small sample sizes and large random effects.

## Results

### Simulation Study Results

Descriptive summaries and insights from each scenario of the simulation study can be seen in *Table 2*. The average error and the total reduction in error for the scenarios can be seen in *Table 3*. For scenario 1, we assumed a relatively uniform spread of variant B across England–

**Table 2. The summary and key insights for the simulation scenarios.**

| Scenario | Scenario summary | Key insights |
|---|---|---|
| 1 | This scenario was designed so that an emerging variant had relatively uniform spatial dispersion across England, with no well-defined clusters of prevalence. Small random effects were present during the data generation process. In several LTLAs there were zero samples, with no LTLA having more than 150 samples. The relatively uniform spatial dispersion would not have been immediately apparent from the data. | In this scenario the measured model performance was high, estimating a spatial spread pattern that was very close to the truth, and it successfully imputed the variant proportion for areas with no samples. |
| 1.1 | This scenario assumed the same spatial dispersion and random effect sizes as scenario 1. However, sample sizes have been reduced so that there were no LTLAs which had more than 25 samples. | Despite small sample sizes at LTLA, the model was able to recover the spatial dispersion pattern successfully. There was some evidence of bias towards a larger proportion of variant B. |
| 1.2 | This scenario assumes the same spatial dispersion and sample sizes as scenario 1. However, the scale of random effects had been increased during the data generating process. | The presence of large random effects impeded model performance. The fitted spatial spread pattern was less smooth than in scenarios 1 and 1.1. |
| 1.3 | This scenario assumed the same spatial dispersion as scenario 1. However, the sample size was reduced and the scale of random effects in the data generating process were increased. | Model performance was again impeded by large random effects, and the fitted spatial dispersion was less smooth than the true spatial dispersion. However, the additional reduction in sample sizes did not lead to a substantial reduction in model performance compared to scenario 1.2. |
| 2 | In this scenario approximately 50% of cases in London were variant B. In LTLAs that were not strongly connected to London the prevalence of variant B was very low. This pattern of geographical concentration in London was observed in real world data of emerging variants. Sample sizes range from 0 to 170 at LTLA level, and random effects were present but small. | The model correctly estimated the spatial dispersion of variant B was concentrated in London, with several emerging hotspots outside of London. |

*(Continued)*

**Table 2.** (Continued)

| Scenario | Scenario summary | Key insights |
|---|---|---|
| 2.1 | This scenario assumed the same spatial dispersion, and random effect sizes as scenario 2. However, sample sizes were reduced so that no LTLA had more than 30 samples. | Despite the small sample sizes, the model correctly estimated that variant B was spatially concentrated in London. However, due to the small sample sizes at LTLA level, some of the emerging hotspots outside of London were not identified. |
| 2.2 | This scenario assumed the same spatial dispersion and sample sizes as scenario 2. However, the scale of the random effects in the data generating process were increased. | The model performed well in terms of estimating the central cluster of cases in London. Additionally, the model did estimate that there were some hotspots outside London, however the spatial dispersion patterns for these hotspots were not well recovered in some instances due to the large random effects. |
| 2.3 | This scenario assumes the same spatial dispersion as scenario 1. However, the sample size has been reduced and the scale of random effects in the data generating process has been increased. | The model performed well at estimating the central cluster of variant B cases in London. Some, but not all, of the hotspots outside of London were also included in the spatial pattern. |
| 3 | In this scenario, we set up several dense clusters of variant B cases. Outside of these clusters, there were very few variant B cases, with the exception of some early spread in London. | The model performed well in terms of estimating the spatial dispersion for each of the localised clusters and captured the initial spread of variant B in London. |
| 3.1 | This scenario assumed the same spatial dispersion and random effect sizes as scenario 3. However, sample sizes were reduced so that no LTLA had more than 25 samples. | The model performed well at estimating the spatial pattern for each of the localised clusters, but the initial spread of variant B in London was not captured due to the small sample sizes at LTLA level. |
| 3.2 | This scenario assumed the same spatial dispersion and sample sizes as scenario 2. However, the scale of random effects in the data generating process were increased. | The model performed well at estimating the spatial dispersion of the clusters outside of London, with some small errors introduced by the random effects. The initial spread of variant B in London was also partially captured, with random effects leading to errors in the estimated spatial patten. |
| 3.3 | This scenario assumed the same spatial dispersion as scenario 3. However, the sample size has been reduced and the scale of the random effects in the data generating process was increased. | The model provided good estimates of the spatial pattern of clusters outside of London and the initial spread within London. In several LTLAs nearby to the clusters the model over-estimated variant B proportion due to the large random effects and small sample sizes at LTLA level. |

**Table 3. The average error for the naïve estimate (%), BYM2 (%), and total reduction in error (%) for the simulation scenarios.**

| Scenario | Average Error for Naïve Estimate (%) | Average Error for BYM2 Estimate (%) | Reduction in Error (%) |
|---|---|---|---|
| 1 | 7.41 | 2.42 | 67.3 |
| 1.1 | 15.11 | 3.48 | 77.0 |
| 1.2 | 10.04 | 5.81 | 42.1 |
| 1.3 | 16.76 | 6.30 | 62.4 |
| 2 | 5.79 | 3.63 | 37.3 |
| 2.1 | 11.03 | 5.20 | 52.9 |
| 2.2 | 8.03 | 5.53 | 31.1 |
| 2.3 | 12.73 | 6.46 | 49.2 |
| 3 | 2.66 | 2.00 | 24.6 |
| 3.1 | 5.50 | 3.21 | 41.6 |
| 3.2 | 3.82 | 2.90 | 24.2 |
| 3.3 | 5.45 | 3.41 | 37.4 |

this was similar to the spread of Omicron BA.5. Sample sizes vary, with some areas having close to 150 sequenced test results, and other areas having far fewer sequenced results. Notably, some areas have no sequenced results, indicated in red within *Fig A1* in the *S1 Text*, and the model must impute the proportion of variant B for these areas. Across all variants of scenario 1, we found that the model offered an improvement over the naïve estimator–particularly when there were very small sample sizes, and small random effects. As the random effects increased in magnitude, the model still offered an improvement over the naïve estimator, however the amount of spatial smoothing performed by the model is reduced as the underlying signal is obscured.

Scenario 2 explores a situation where a new variant had emerged and was concentrated in London and a few other regions that were highly connected to London. This was designed to approximate the early spread of Omicron BA.1, where it clearly reached large well connected population centres first, such as London, Manchester, Birmingham, before rapidly spreading outwards. For the baseline version of the scenario, with small random effects and large sample sizes, we found both the BYM2 and the naïve estimator were able to produce good estimates of the localised spread patterns with the BYM2 modelling offering a modest improvement over the naïve estimator. However, the benefit of the BYM2 modelling is highlighted for the scenarios where there were small sample sizes and large random effects, where the modelling offered a significant improvement over the naïve estimator.

For the final scenario, we considered small highly localised outbreaks of variant B. This scenario may occur, if a new highly infectious variant evolved in England, and the initial infections were geographically spread out. This scenario is important, given that highly localised clusters of infections can be subject to targeted interventions such as enhanced contact tracing. For our simulated scenario, variant B was localised in the West Midlands, Essex, parts of

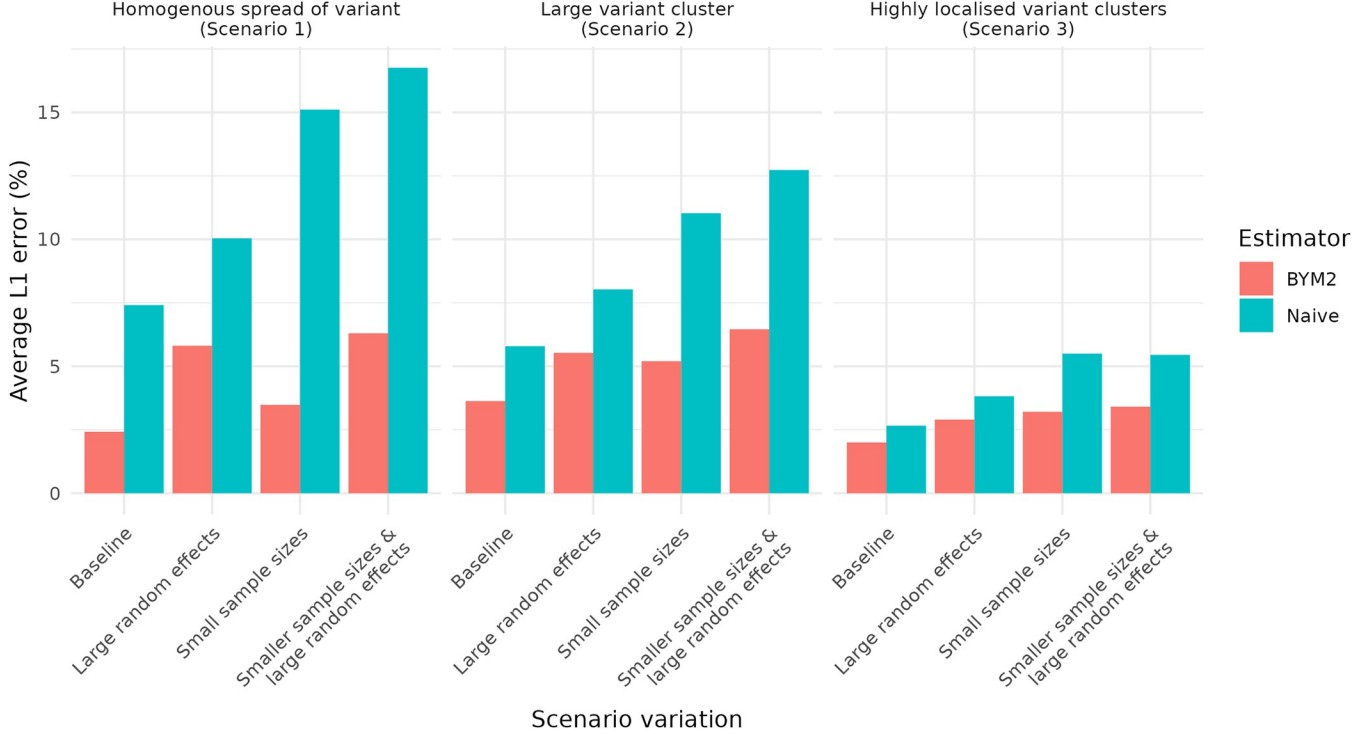

**Fig 2. A clustered bar chart of the reduction in the average L1 error, for each scenario, variation, and model.**

central London, and Gloucestershire. We found that the performance of BYM2 modelling overall provided improved estimates over the naïve estimator for all version of this scenario, with the BYM2 modelling being particularly important when sample sizes were particularly small. The BYM2 modelling did have slight difficulty in estimating the early spread of the variant in areas of central London, where the weak signal is possibly obscured by random effects or noise.

Overall, we concluded that for all simulated scenarios, there was no discernible disadvantages to using this model framework to estimate the spatial spread of a new variant. When the sample sizes we provided to the model were particularly small, it became significantly more important to use BYM2 model to estimate the spatial spread of variant and to produce outputs that could provide adequate situational awareness of the spread pattern. The scenarios with large random effects showed a reduction in model performance, however there was still an improvement over the naïve estimator. The magnitude of the random effects we considered in our model may be unrealistic, given what has been generally observed in real-world data, however the purpose was to stress test the model to understand its performance in difficult conditions that may arise as the SARS-CoV-2 response continues to evolve. The average L1 error across the modelled scenarios can be seen in *Fig 2* and the full results can be seen in *Figs A1-A36* in the *S1 Text*.

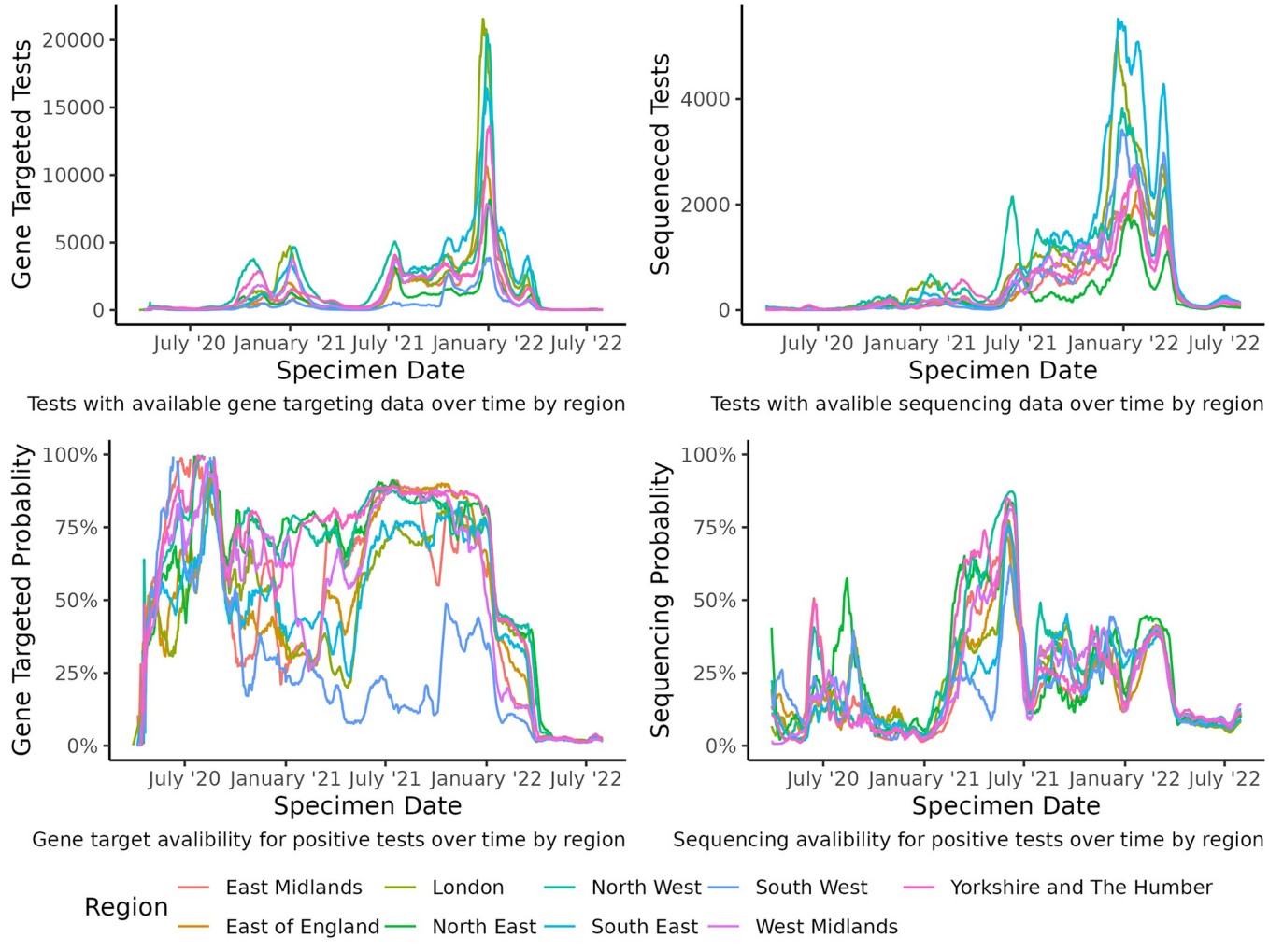

**Fig 3. Gene target and sequencing data coverage over time for RT-PCR positive tests in England.**

## Geographic coverage of sequenced RT-PCR tests and RT-PCR tests with gene targets

The spatio-temporal variation of sequenced, genotyped and RT-PCR tests that provided gene target information across the regions of England can be seen in *Fig 3*. We found that overall, the South West had the lowest rate of tests that provided gene targets and Yorkshire and the Humber, the North West and the North East had the greatest proportion. The probability of a test being sequenced or genotyped was highly temporally heterogeneous with the greatest proportion of tests sequenced observed in 2021. We also found, akin to gene target test coverage, that the South West had the lowest proportion of RT-PCR tests that were sequenced and that the regions in the North of England had the greatest proportion. Since the end of mass testing in April 2022 all regions experienced a substantial decline in the proportion of tests being sequenced but that the level is now roughly constant. Tests that provide information on the gene targets of a case has largely ceased since April 2022.

## Spatial modelling of emerging variants

Resurgent epidemic waves of SARS-CoV-2 incidence have been largely a consequence of emerging variants in England. The BYM2 spatial modelling estimates for the growth of the Alpha, Delta, Omicron BA.1, Omicron BA.2, and Omicron BA.5, through the use of sequenced and genotyped RT-PCR test data, can be seen in *Figs A37* to *A86* in the *S1 Text*. The modelled estimates were influenced by the spatial dispersion of the variant, localised rates of growth, sequencing coverage, the number of tests conducted and the temporal spacing between them.

There was increased sequencing coverage in the South East and London in November 2020 due to the detection of the Alpha variant (*Fig A37* in the *S1 Text*). During the early phases of growth for Alpha it was concentrated in Kent and the associated networks of this county in the South East and London region of England (in particular, Norfolk, London boroughs, East Sussex, and Essex), which can be seen in *Fig 4*. The dissemination of Alpha, as a result, came through the branching out of networks associated with these areas. The Alpha wave remained predominantly concentrated in the London and South East regions of England, until a rapid increase in the spatial dispersion of this variant across all regions when it reached, nationally, around 60% of sequenced cases (*Fig A43* in the *S1 Text*).

In *Figs A47* to *A56* in the *S1 Text* we found that the proportion of PCR tests that were sequenced markedly increased on targeted locations where Delta was detected, which can cause increased geographic bias in the naïve test positivity data. In contrast to Alpha, with its high early concentration in the South East and London regions of England, the Delta variant saw more dispersive clustering across LTLAs in England *(Fig 5)*, though particularly in the locations within the North West, London, and South East regions. This led to a different spatially dispersive pattern relative to Alpha and when Delta was at a prevalence of around 60% nationally (*Fig* A53 in the *S1 Text*), it was highly concentrated in areas connecting the North West and the South East of England, with lower prevalence in the South West and North East.

The early phases of growth for Omicron BA.1 were predominantly concentrated in the South East, London, and the North West (*Fig 6*). This is a similar to the spatial pattern observed for Delta, however the early growth phase of Delta was characterised by smaller clusters of LTLAs with higher prevalence, due to NPIs, whereas Omicron BA.1 was more spatially dispersive within these regions. During the early growth phases of Omicron BA.1 increased sequencing was conducted within targeted LTLAs, and this period had greatest absolute number of tests that were sequenced or genotyped in England. The spatial spread of Omicron BA.1

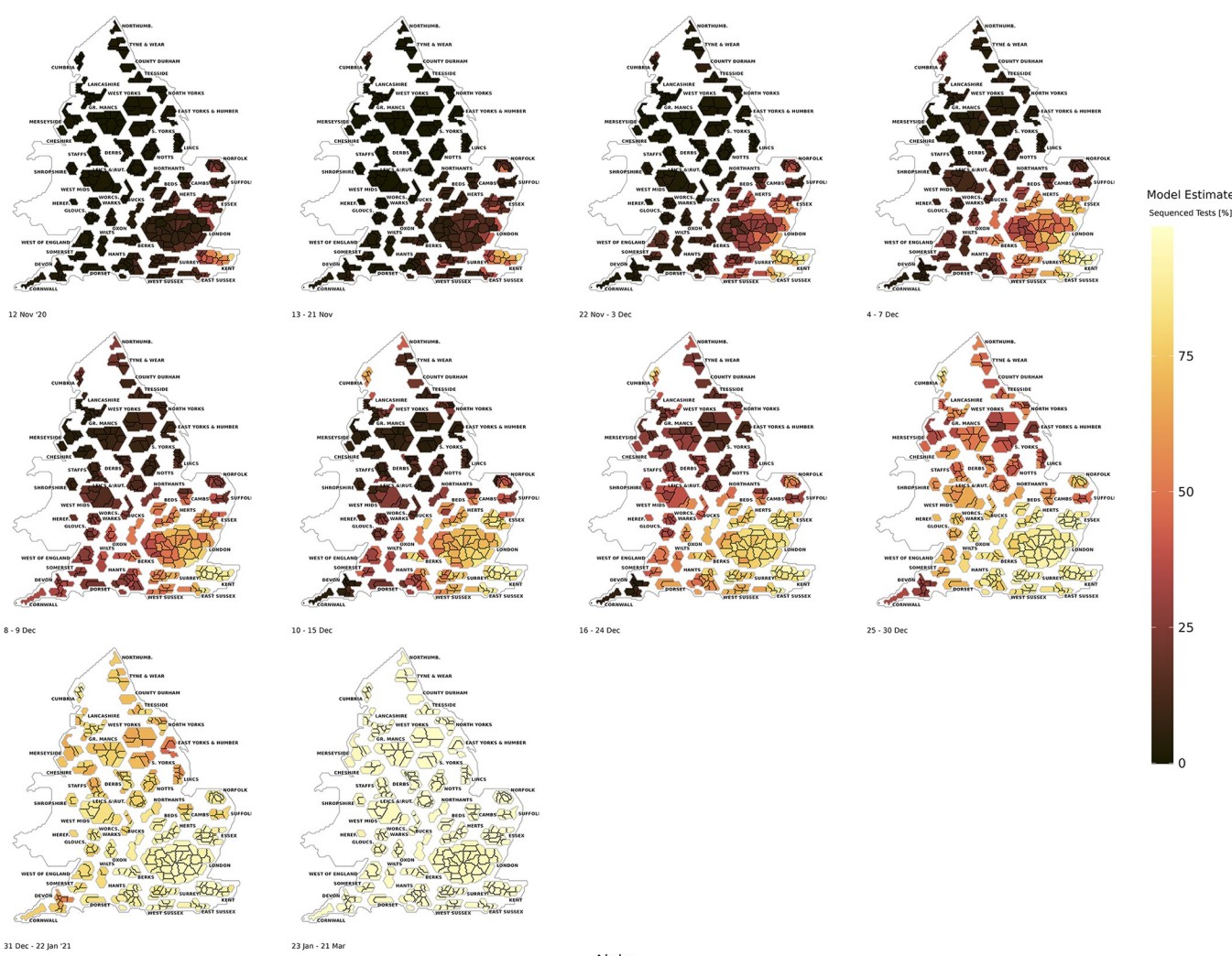

**Fig 4. The BYM2 estimated model positivity of the Alpha variant as a proportion of sequenced tests from 12th November 2020 to 21st March 2022.** Created from geographical files using the House of Commons Library, which is under the Open Parliament License v3.0.

can be seen through its propagation across the mobility networks closely connected to London and South East regions.

Omicron BA.2 began to replace Omicron BA.1 in February 2022 (*Figs A67 to A76 in the S1 Text*). When Omicron BA.2 began to grow it was more spatially dispersed and regionally homogenous than had been observed for previously dominant variants. This was likely a consequence of earlier introductions of this variant in late 2021. However, the modelling illustrated that this variant was still predominantly dispersed from the London and South East regions in the early stages of growth (*Fig 7*). We found that sequencing was also conducted in a less targeted manner during the growth phases of Omicron BA.2 than had been observed in December 2021 and January 2022 to detect Omicron BA.1.

The end of free mass testing occurred during the final stages of the Omicron BA.2 growth period and therefore, the spatial coverage reduced substantially during the growth in Omicron BA.5. Omicron BA.5 began to grow in May 2022 and this variant was predominantly in competition with Omicron BA.4 in the early growth phases. We found Omicron BA.5 (*Fig 8*) was

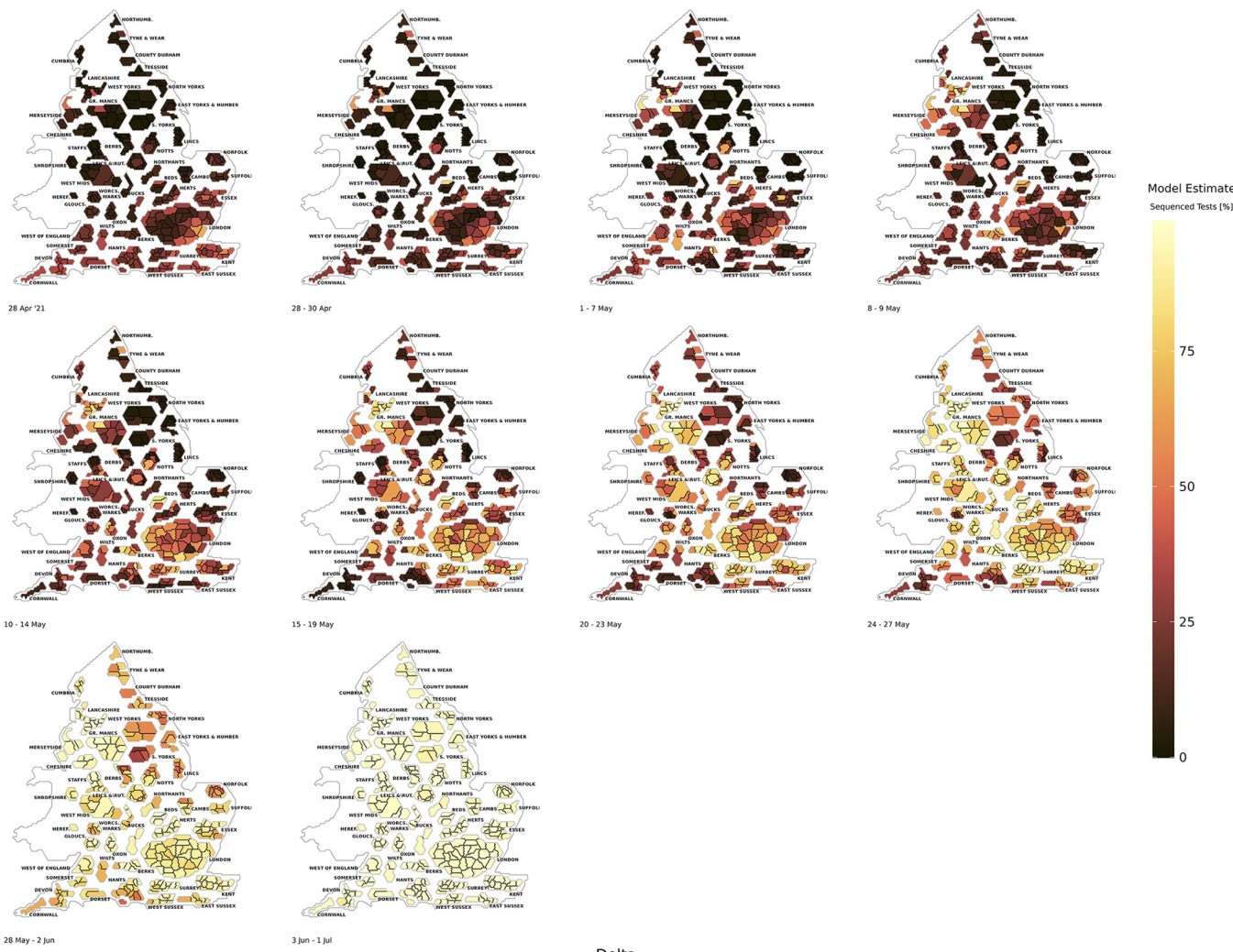

**Fig 5. The BYM2 estimated model positivity of the Delta variant as a proportion of sequenced tests from 28th April 2021 to 1st July 2021.** Created from geographical files using the House of Commons Library, which is under the Open Parliament License v3.0.

notably more spatially dispersed across England in the early phases of growth compared to Alpha, Delta, and Omicron BA.1, which was similar to the pattern observed for Omicron BA.2.

The more recent waves of incidence have been difficult to ascribe to one particular lineage due to the considerable diversity now observed for Omicron sublineages. We found that since June 2022 there have been growth and considerable dominance of variants that share convergent mutations in the RBD (*Fig A87* in the *S1 Text*). This has been relatively spatially homogenous with particular concentrations around London, the South East, the North West, and the Midlands regions. We also found considerable spatial dispersion for three significant RBD mutations F486V, N460K, and R346T in *Figs A88* to *A90* in the *S1 Text*.

## Discussion

The distinctive and heterogeneous dispersion of novel variants of SARS-CoV-2 has confounded efforts to understand and slow their growth. This has been impacted by spatio-

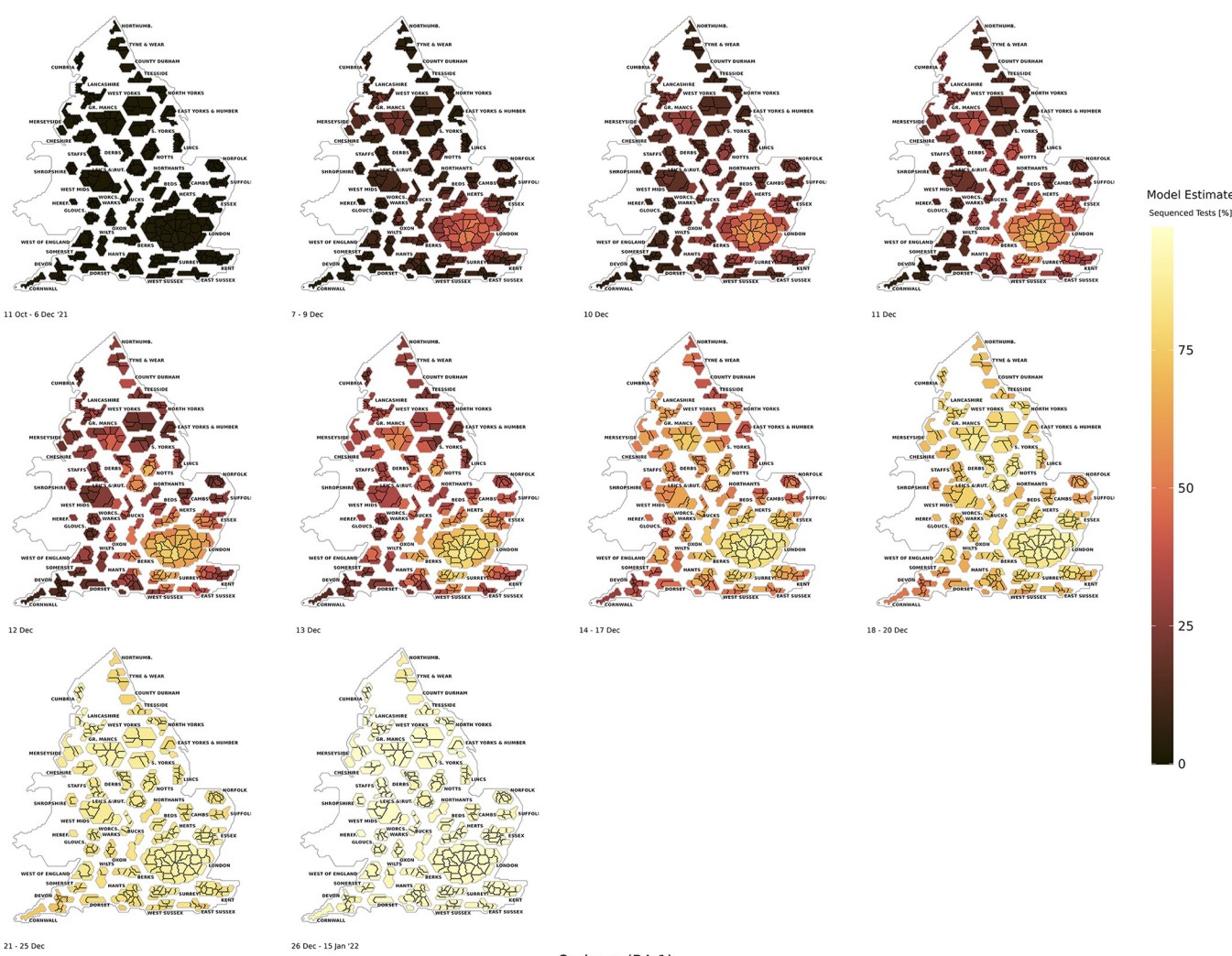

**Fig 6. The BYM2 estimated model positivity of the Omicron BA.1 variant as a proportion of sequenced tests from 11th October 2021 to 15th January 2022.** Created from geographical files using the House of Commons Library, which is under the Open Parliament License v3.0.

temporal variation of testing, sequencing, and genotyping coverage across the pandemic. We found that through the development of a mobility derived network with a Bayesian BYM2 model we could substantially improve the spatial understanding of dispersive variant growth. We have assessed this approach through simulation analysis to measure model performance and real world data to understand the spatial growth of emerging variants. The model was found to be of most utility when sample sizes are small, as it can provide improved situational public health awareness of the spatial pattern of an emerging variant. This has become of increased significance with the cessation of free mass testing in the UK and the reduction in variant identification through RT-PCR tests.

Across the SARS-CoV-2 pandemic, probable and confirmed novel variants were identified through whole-genome sequencing and genotyped RT-PCR tests. In closer to real time, emerging variants were also identified through RT-PCR tests that provided gene targets. We found considerable geographical heterogeneity observed in the probability that a given test is either sequenced or has gene target information available [14]. There is a considerable time lag

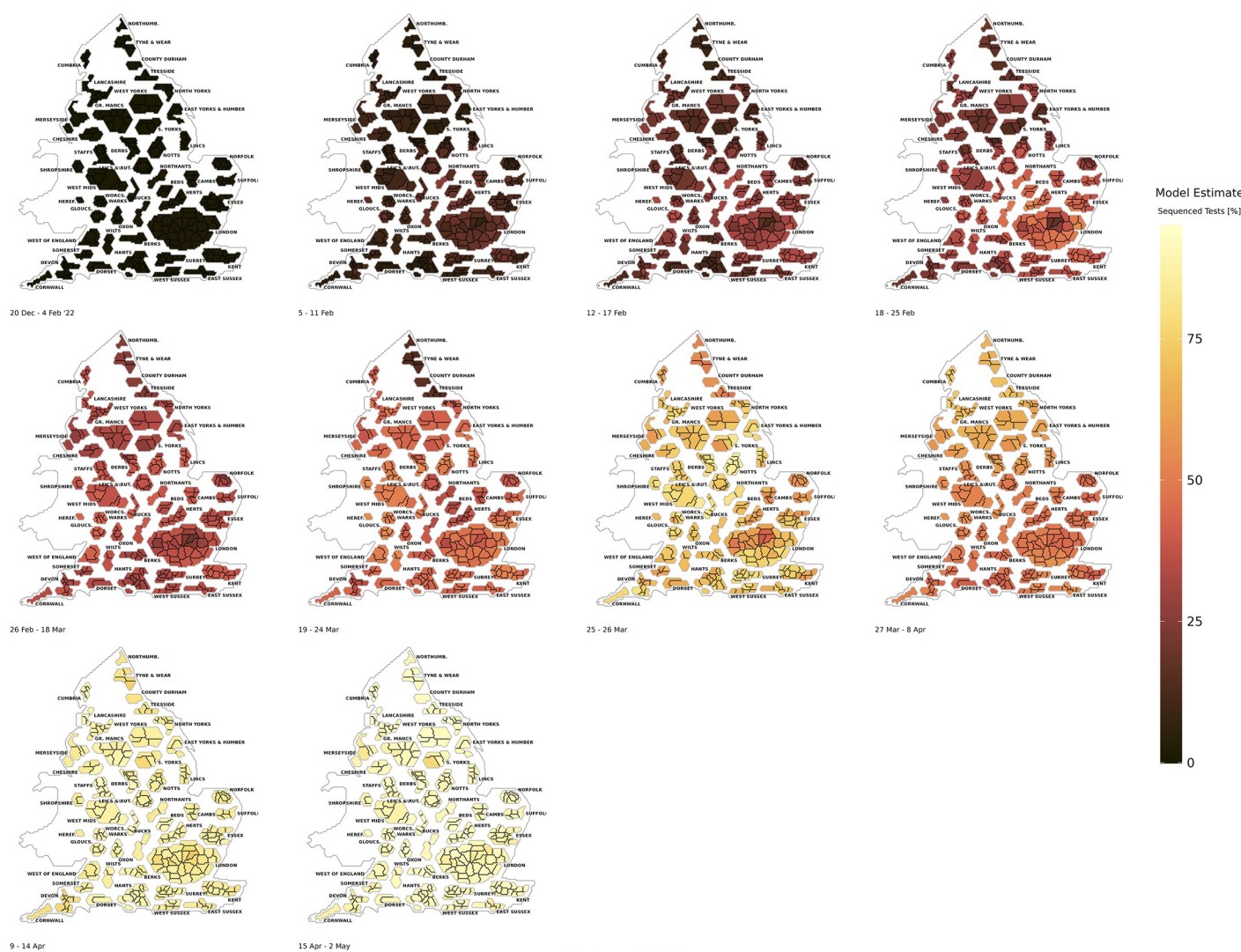

**Fig 7. The BYM2 estimated model positivity of the Omicron BA.2 variant as a proportion of sequenced tests from 20th December 2021 to 2nd May 2022.** Created from geographical files using the House of Commons Library, which is under the Open Parliament License v3.0.

for whole genome sequencing, of around 14 days, that can delay the understanding of variant spatial patterns of growth in real time [37]. Tests that provide gene targets were provided in a far more timely fashion but were a cruder approximation of the variant under investigation. Alpha and Omicron BA.1 were identified through S-gene target failure [14,17] with Delta and Omicron BA.2 being S-gene target positive variants. However, a proportion of these tests are incorrectly designated, and this was found to be exacerbated with CT values over 30 [38]. Moreover, gene targets were only available for RT-PCR tests in certain geographic locations in the UK and therefore some areas were preferentially sampled. We found that the model performance of the BYM2 with a mobility derived network was most pronounced in scenarios where this type of sampling bias was apparent, particularly when samples were small and missing within certain locations.

The spatial patterns for novel variants were a consequence of the number of seeding events, the localised susceptibility of population, the extent of NPIs and behavioural change over the

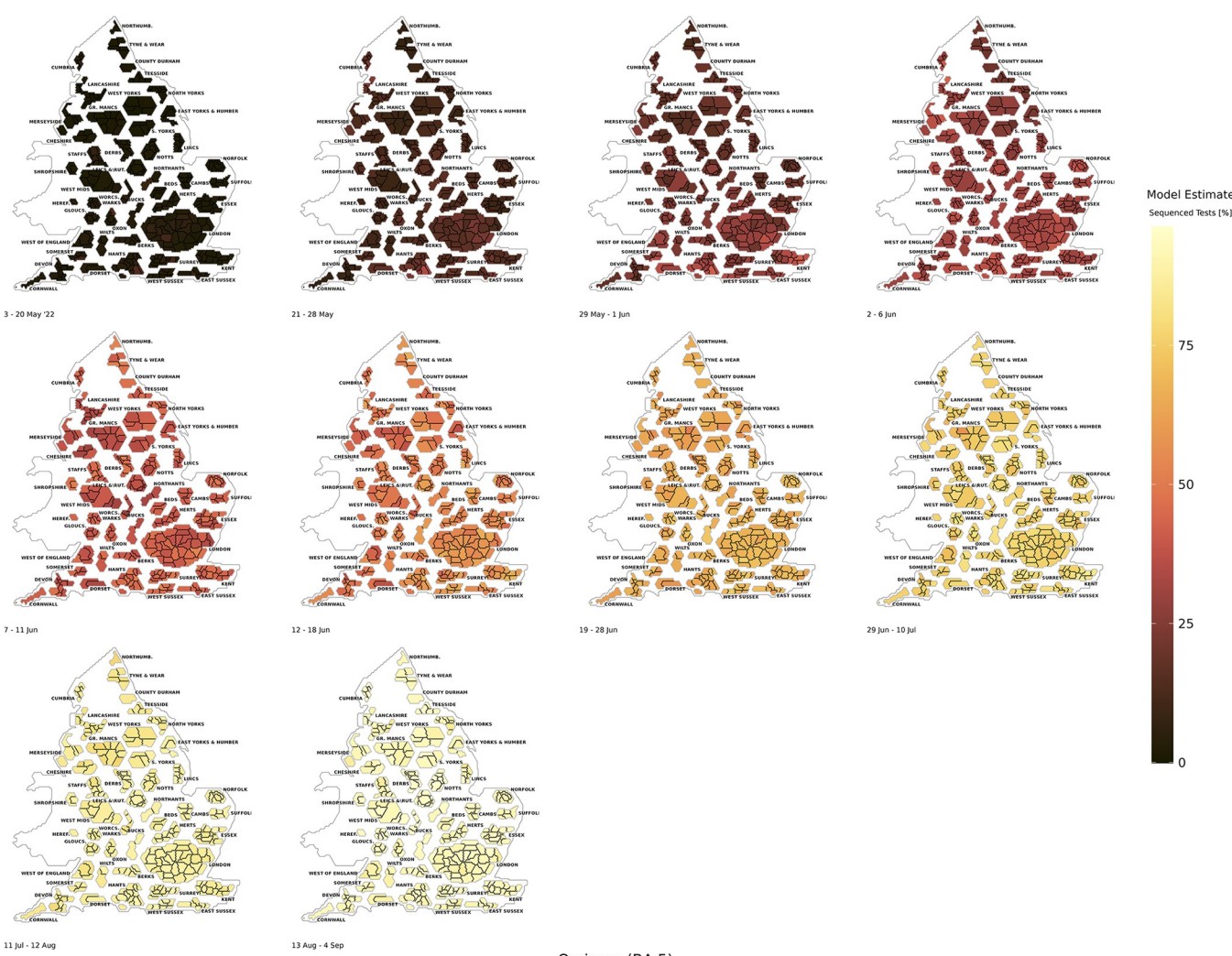

**Fig 8. The BYM2 estimated model positivity of the Omicron BA.5 variant as a proportion of sequenced tests from 31st May 2022 to 4th September 2022.** Created from geographical files using the House of Commons Library, which is under the Open Parliament License v3.0.

pandemic. We found the increased connectivity of the London region plays an important role in the mobility network of England. Every variant of SARS-CoV-2 that became dominant saw early clustering around the London region and its dispersive mobility networks drove early increases in transmission. This is particularly apparent for Alpha, which was initially detected in Kent [13], and due to its close proximity and connectedness to the London region it spread across the South East region very quickly. The significance of the London region is also due to increased international travel arriving through this region and therefore, a higher concentration of importations of novel variants. The introduction of Delta was reliant upon international importation routes, which concentrated in the North West and London. Moreover, transmission of Omicron BA.1 was initially heavily concentrated in the South East and London, which had the greatest international links to South Africa. Omicron BA.2 and BA.5 were less dependent upon the London network for their spatial dissemination. This is a consequence of the cessation of NPIs and resumption of more normative behavioural patterns, which we found increased the strength of connectivity across the country.

We investigated the idea of convergent evolution in RBD mutations that may aid in the immunological evasion from previous variants or vaccination [39]. Despite many of these variants deriving from divergent evolutionary backgrounds they have acquired RBD mutations that are convergent. Furthermore, there is evidence that mutations on these residues are evasive of prevailing antibody responses [40]. This was found most strikingly in sublineages of BA.2.75, BA.5 and XBB variants, which have showed considerable growth rate advantages [41]. This significant increase in variant diversity, with similar fitness advantages, has made tracking individual variants more difficult for operational public health response. The September 2022 wave of SARS-CoV-2 incidence was related to an increase in variants that share common RBD mutations. Therefore, understanding the spatial dispersion of variants that share mutations, which may confer a transmission advantage, has become of greater public health relevance.

## Conclusion

Understanding the spatial spread of emerging variants is essential to be able to provide an adequate public health response. The cessation in free mass testing in the UK has substantially reduced the geographic coverage and the sample size of RT-PCR tests. Consequently, it has become increasingly difficult to understand the spatial dispersion of emerging SARS-CoV-2 variants through test positivity. Through simulation scenario modelling we found the BYM2 model framework proposed in this paper was able to substantially improve the understanding of a variant's spatial dispersion relative to unadjusted RT-PCR positivity. We found that when the sample sizes we provided to the model were particularly small or testing coverage was sparse, it became significantly more important to use this approach to estimate the spatial spread of a variant. The simulation scenarios which included large random effects were found to reduce the performance of the model, however there was still an improvement over the naïve estimator. The development of this proposed modelling framework can help to improve real-time public health situational awareness during outbreaks of novel variants of SARS--CoV-2.

### Ethics statement

This study was conducted for the purpose of informing the outbreak response to the COVID-19 pandemic. Work was undertaken in line with national data regulations. It only employed and accessed fully anonymised population level data from UKHSA in a secure research.

## Supporting information

**S1 Text.** Fig A1. Comparisons of the estimates from the Naive estimator, and the BYM2. Fig A2. Plots of the true value against the fitted value for the Naive estimator and BYM2. Fig A3. ROC curve of L1 error for the Naive estimate and BYM2. Fig A4. Comparisons of the estimates from the Naive estimator, and the BYM2. Fig A5. Plots of the true value against the fitted value for the Naive estimator and BYM2. Fig A6. ROC curve of L1 error for the Naive estimate and BYM2. Fig A7. Comparisons of the estimates from the Naive estimator, and the BYM2. Fig A8. Plots of the true value against the fitted value for the Naive estimator and BYM2. Fig A9. ROC curve of L1 error for the Naive estimate and BYM2. Fig A10. Comparisons of the estimates from the Naive estimator, and the BYM2. Fig A11. Plots of the true value against the fitted value for the Naive estimator and BYM2. Fig A12. ROC curve of L1 error for the Naive estimate and BYM2. Fig A13. Comparisons of the estimates from the Naive estimator, and the BYM2. Fig A14. Plots of the true value against the fitted value for the Naive estimator and BYM2. Fig A15. ROC curve of L1 error for the Naive estimate and BYM2. Fig A16.

Comparisons of the estimates from the Naive estimator, and the BYM2. Fig A17. Plots of the true value against the fitted value for the Naive estimator and BYM2. Fig A18. ROC curve of L1 error for the Naive estimate and BYM2. Fig A19. Comparisons of the estimates from the Naive estimator, and the BYM2. Fig A20. Plots of the true value against the fitted value for the Naive estimator and BYM2. Fig A21. ROC curve of L1 error for the Naive estimate and BYM2. Fig A22. Comparisons of the estimates from the Naive estimator, and the BYM2. Fig A23. Plots of the true value against the fitted value for the Naive estimator and BYM2. Fig A24. ROC curve of L1 error for the Naive estimate and BYM2. Fig A25. Comparisons of the estimates from the Naive estimator, and the BYM2. Fig A26. Plots of the true value against the fitted value for the Naive estimator and BYM2. Fig A27. ROC curve of L1 error for the Naive estimate and BYM2. Fig A28. Comparisons of the estimates from the Naive estimator, and the BYM2. Fig A29. Plots of the true value against the fitted value for the Naive estimator and BYM2. Fig A30. ROC curve of L1 error for the Naive estimate and BYM2. Fig A31. Comparisons of the estimates from the Naive estimator, and the BYM2. Fig A32. Plots of the true value against the fitted value for the Naive estimator and BYM2. Fig A33. ROC curve of L1 error for the Naive estimate and BYM2. Fig A34. Comparisons of the estimates from the Naive estimator, and the BYM2. Fig A35. Plots of the true value against the fitted value for the Naive estimator and BYM2. Fig A36. ROC curve of L1 error for the Naive estimate and BYM2. Fig A37. The BYM2 estimated model positivity of the Alpha variant as a proportion of sequenced tests, the model adjustment, the proportion of tests that were sequenced, and the sample size for the time period. Fig A38. The BYM2 estimated model positivity of the Alpha variant as a proportion of sequenced tests, the model adjustment, the proportion of tests that were sequenced, and the sample size for the time period. Fig A39. The BYM2 estimated model positivity of the Alpha variant as a proportion of sequenced tests, the model adjustment, the proportion of tests that were sequenced, and the sample size for the time period. Fig A40. The BYM2 estimated model positivity of the Alpha variant as a proportion of sequenced tests, the model adjustment, the proportion of tests that were sequenced, and the sample size for the time period. Fig A41. The BYM2 estimated model positivity of the Alpha variant as a proportion of sequenced tests, the model adjustment, the proportion of tests that were sequenced, and the sample size for the time period. Fig A42. The BYM2 estimated model positivity of the Alpha variant as a proportion of sequenced tests, the model adjustment, the proportion of tests that were sequenced, and the sample size for the time period. Fig A43. The BYM2 estimated model positivity of the Alpha variant as a proportion of sequenced tests, the model adjustment, the proportion of tests that were sequenced, and the sample size for the time period. Fig A44. The BYM2 estimated model positivity of the Alpha variant as a proportion of sequenced tests, the model adjustment, the proportion of tests that were sequenced, and the sample size for the time period. Fig A45. The BYM2 estimated model positivity of the Alpha variant as a proportion of sequenced tests, the model adjustment, the proportion of tests that were sequenced, and the sample size for the time period. Fig A46. The BYM2 estimated model positivity of the Alpha variant as a proportion of sequenced tests, the model adjustment, the proportion of tests that were sequenced, and the sample size for the time period. Fig A47. The BYM2 estimated model positivity of the Delta variant as a proportion of sequenced tests, the model adjustment, the proportion of tests that were sequenced, and the sample size for the time period. Fig A48. The BYM2 estimated model positivity of the Delta variant as a proportion of sequenced tests, the model adjustment, the proportion of tests that were sequenced, and the sample size for the time period. Fig A49. The BYM2 estimated model positivity of the Delta variant as a proportion of sequenced tests, the model adjustment, the proportion of tests that were sequenced, and the sample size for the time period. Fig A50. The BYM2 estimated model positivity of the Delta variant as a proportion of sequenced tests, the model adjustment, the proportion of tests that were sequenced,

and the sample size for the time period. Fig A51. The BYM2 estimated model positivity of the Delta variant as a proportion of sequenced tests, the model adjustment, the proportion of tests that were sequenced, and the sample size for the time period. Fig A52. The BYM2 estimated model positivity of the Delta variant as a proportion of sequenced tests, the model adjustment, the proportion of tests that were sequenced, and the sample size for the time period. Fig A53. The BYM2 estimated model positivity of the Delta variant as a proportion of sequenced tests, the model adjustment, the proportion of tests that were sequenced, and the sample size for the time period. Fig A54. The BYM2 estimated model positivity of the Delta variant as a proportion of sequenced tests, the model adjustment, the proportion of tests that were sequenced, and the sample size for the time period. Fig A55. The BYM2 estimated model positivity of the Delta variant as a proportion of sequenced tests, the model adjustment, the proportion of tests that were sequenced, and the sample size for the time period. Fig A56. The BYM2 estimated model positivity of the Delta variant as a proportion of sequenced tests, the model adjustment, the proportion of tests that were sequenced, and the sample size for the time period. Fig A57. The BYM2 estimated model positivity of the Omicron BA.1 variant as a proportion of sequenced tests, the model adjustment, the proportion of tests that were sequenced, and the sample size for the time period. Fig A58. The BYM2 estimated model positivity of the Omicron BA.1 variant as a proportion of sequenced tests, the model adjustment, the proportion of tests that were sequenced, and the sample size for the time period. Fig A59. The BYM2 estimated model positivity of the Omicron BA.1 variant as a proportion of sequenced tests, the model adjustment, the proportion of tests that were sequenced, and the sample size for the time period. Fig A60. The BYM2 estimated model positivity of the Omicron BA.1 variant as a proportion of sequenced tests, the model adjustment, the proportion of tests that were sequenced, and the sample size for the time period. Fig A61. The BYM2 estimated model positivity of the Omicron BA.1 variant as a proportion of sequenced tests, the model adjustment, the proportion of tests that were sequenced, and the sample size for the time period. Fig A62. The BYM2 estimated model positivity of the Omicron BA.1 variant as a proportion of sequenced tests, the model adjustment, the proportion of tests that were sequenced, and the sample size for the time period. Fig A63. The BYM2 estimated model positivity of the Omicron BA.1 variant as a proportion of sequenced tests, the model adjustment, the proportion of tests that were sequenced, and the sample size for the time period. Fig A64. The BYM2 estimated model positivity of the Omicron BA.1 variant as a proportion of sequenced tests, the model adjustment, the proportion of tests that were sequenced, and the sample size for the time period. Fig A65. The BYM2 estimated model positivity of the Omicron BA.1 variant as a proportion of sequenced tests, the model adjustment, the proportion of tests that were sequenced, and the sample size for the time period. Fig A66. The BYM2 estimated model positivity of the Omicron BA.1 variant as a proportion of sequenced tests, the model adjustment, the proportion of tests that were sequenced, and the sample size for the time period. Fig A67. The BYM2 estimated model positivity of the Omicron BA.2 variant as a proportion of sequenced tests, the model adjustment, the proportion of tests that were sequenced, and the sample size for the time period. Fig A68. The BYM2 estimated model positivity of the Omicron BA.2 variant as a proportion of sequenced tests, the model adjustment, the proportion of tests that were sequenced, and the sample size for the time period. Fig A69. The BYM2 estimated model positivity of the Omicron BA.2 variant as a proportion of sequenced tests, the model adjustment, the proportion of tests that were sequenced, and the sample size for the time period. Fig A70. The BYM2 estimated model positivity of the Omicron BA.2 variant as a proportion of sequenced tests, the model adjustment, the proportion of tests that were sequenced, and the sample size for the time period. Fig A71. The BYM2 estimated model positivity of the Omicron BA.2 variant as a proportion of sequenced tests, the model adjustment, the proportion of tests that were

sequenced, and the sample size for the time period. Fig A72. The BYM2 estimated model positivity of the Omicron BA.2 variant as a proportion of sequenced tests, the model adjustment, the proportion of tests that were sequenced, and the sample size for the time period. Fig A73. The BYM2 estimated model positivity of the Omicron BA.2 variant as a proportion of sequenced tests, the model adjustment, the proportion of tests that were sequenced, and the sample size for the time period. Fig A74. The BYM2 estimated model positivity of the Omicron BA.2 variant as a proportion of sequenced tests, the model adjustment, the proportion of tests that were sequenced, and the sample size for the time period. Fig A75. The BYM2 estimated model positivity of the Omicron BA.2 variant as a proportion of sequenced tests, the model adjustment, the proportion of tests that were sequenced, and the sample size for the time period. Fig A76. The BYM2 estimated model positivity of the Omicron BA.2 variant as a proportion of sequenced tests, the model adjustment, the proportion of tests that were sequenced, and the sample size for the time period. Fig A77. The BYM2 estimated model positivity of the Omicron BA.5 variant as a proportion of sequenced tests, the model adjustment, the proportion of tests that were sequenced, and the sample size for the time period. Fig A78. The BYM2 estimated model positivity of the Omicron BA.5 variant as a proportion of sequenced tests, the model adjustment, the proportion of tests that were sequenced, and the sample size for the time period. Fig A79. The BYM2 estimated model positivity of the Omicron BA.5 variant as a proportion of sequenced tests, the model adjustment, the proportion of tests that were sequenced, and the sample size for the time period. Fig A80. The BYM2 estimated model positivity of the Omicron BA.5 variant as a proportion of sequenced tests, the model adjustment, the proportion of tests that were sequenced, and the sample size for the time period. Fig A81. The BYM2 estimated model positivity of the Omicron BA.5 variant as a proportion of sequenced tests, the model adjustment, the proportion of tests that were sequenced, and the sample size for the time period. Fig A82. The BYM2 estimated model positivity of the Omicron BA.5 variant as a proportion of sequenced tests, the model adjustment, the proportion of tests that were sequenced, and the sample size for the time period. Fig A83. The BYM2 estimated model positivity of the Omicron BA.5 variant as a proportion of sequenced tests, the model adjustment, the proportion of tests that were sequenced, and the sample size for the time period. Fig A84. The BYM2 estimated model positivity of the Omicron BA.5 variant as a proportion of sequenced tests, the model adjustment, the proportion of tests that were sequenced, and the sample size for the time period. Fig A85. The BYM2 estimated model positivity of the Omicron BA.5 variant as a proportion of sequenced tests, the model adjustment, the proportion of tests that were sequenced, and the sample size for the time period. Fig A86. The BYM2 estimated model positivity of the Omicron BA.5 variant as a proportion of sequenced tests, the model adjustment, the proportion of tests that were sequenced, and the sample size for the time period. Fig A87. The BYM2 estimated model positivity of convergent receptor binding domain mutations as a proportion of sequenced tests from 24th June 2022 to 27th October 2022. Fig A88. The BYM2 estimated model positivity of F486V receptor binding domain mutation as a proportion of sequenced tests from 24th June 2022 to 27th October 2022. Fig A89. The BYM2 estimated model positivity of N460K receptor binding domain mutation as a proportion of sequenced tests from 24th June 2022 to 27th October 2022. Fig A90. The BYM2 estimated model positivity of R346T receptor binding domain mutation as a proportion of sequenced tests from 24th June 2022 to 27th October 2022.
(DOCX)

**S1 Data. The Stan code of the Bayesian BYM2 model structure.**
(ZIP)

## Author Contributions

**Conceptualization:** Thomas Ward, Martyn Fyles.

**Data curation:** Thomas Ward, William Ferguson, Martyn Fyles.

**Formal analysis:** Thomas Ward, William Ferguson, Martyn Fyles.

**Investigation:** Thomas Ward, Mitzi Morris, Bob Carpenter, Christopher Overton, Martyn Fyles.

**Methodology:** Thomas Ward, Mitzi Morris, Andrew Gelman, Bob Carpenter, Christopher Overton, Martyn Fyles.

**Project administration:** Thomas Ward.

**Supervision:** Thomas Ward.

**Validation:** Thomas Ward.

**Visualization:** Thomas Ward, William Ferguson, Martyn Fyles.

**Writing – original draft:** Thomas Ward, Andrew Gelman, Bob Carpenter, Christopher Overton, Martyn Fyles.

**Writing – review & editing:** Thomas Ward, Martyn Fyles.

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
