## [Decision Letter · Decision Letter 0]

24 Apr 2023

Dear Mr Ward,

Thank you very much for submitting your manuscript "Bayesian Spatial Modelling of Localised SARS-CoV-2 Transmission through Mobility Networks in the United Kingdom" for consideration at PLOS Computational Biology. As with all papers reviewed by the journal, your manuscript was reviewed by members of the editorial board and by several independent reviewers. The reviewers appreciated the attention to an important topic. Based on the reviews, we are likely to accept this manuscript for publication, providing that you modify the manuscript according to the review recommendations.

Sincerely,

Joseph T. Wu

Academic Editor

PLOS Computational Biology

Virginia Pitzer

Section Editor

PLOS Computational Biology

Reviewer's Responses to Questions

**Comments to the Authors:**

Reviewer #1: Many thanks for the excellent study. This tackles an important research question. Mostly the article is clear, but i have the following comments:

1) line 58: "We have then spatially modelled replacing variant growth through gene targets, whole-genome sequencing, and genotyped RT-PCR tests" doesn't quite read well, suggest a slight change in structure

2) line 178: the subscript k is introduced but not explicitly defined to be over spatial unit, this is only implied and may benefit from clarity.

3) about line 200: the authors present the force of infection from other patches in the meta-population but did they consider impact of diurnal flux due to commuting (night-time/daytime mixing is different). would this impact on results?

4) Results: it would be helpful to have a table of parameters being fitted with the prior/posterior estimates in the results. Consider % L1 error requires a little thinking about to the lay/general reader. This may be in SI but I couldn't open the SI in the pdf available to conduct review.

Reviewer #2: The authors have presented a method for modeling viral transmission using mobility data. The paper is well polished and their use of scenario-based simulation studies to validate their method is commendable. The methodology is well explained and I was able to follow it even as someone with a limited mathematical background. I think this paper is well suited for PLOS Computational Biology, but it still needs some more work.

Major concern: this is far from the first study on incorporating mobility data into viral transmission models. And yet it is unclear if/how the BYM2 model differs from other methods at achieving this task. I would like to see how this method compares to previous approaches.

Code availability: Perhaps I missed it somewhere, but if not I encourage the authors to make their code open source and publicly available. Even if messy, it can still be repurposed by researchers who may want to apply the methods to other parts of the world

General comment: is this study about England or the UK? The title and body of the article suggests the UK, but all of the maps are of England and there is no mention of Scotland, Ireland, or Wales. Perhaps it’s too much of a pain to change the title of the paper at this stage, but unless I’m missing something, it should be explicit that this article is about England and not the UK.

General comment: the maps (esp Fig 4-6) are very low resolution and the labels are difficult to read even when zoomed in

General comment: from an evolutionary perspective I’m not so keen on the term ‘replacement’, which is used throughout the article. My problem with the term replacement is that the trend is usually variant 1 infects X people per day, and then variant 2 comes along which infects >>X people per day, thereby making variant 1 appear obsolete. The new variant does not simply replace the old one, it also supersedes it. I suggest the term replacement should itself be replaced to better capture underlying evolutionary processes, eg. succeed, supersede, outcompete, outgrow.

General comment: where are the prior distributions? This is a Bayesian analysis, so the priors should be explicated somewhere, perhaps in the SI.

Line 3: Here variants of concern are introduced, but on line 89 the article refers to variants of interest. VOC and VOI have different definitions according to the WHO, so it should be clear somewhere if this work and its methodologies make the distinction. Or perhaps use the phrase ‘emerging variants’ where applicable to avoid the need to define these terms

Lower tier local authorities -> as a non-Brit I am not familiar with this term, I gather it means district councils? Could you provide a simple description/example of what an LTLA is so the reader does not need to google it

Line 21: subsequently -> consequently

Line 27: “Transmission of this variant… South of England.” -> statement should be supported by a reference

Line 63: “ethical approval” should start with a new sentence

Line 88: “of which y_i=1 indicate the tested individual was infected with our target variant of interest” -> I think this is erroneous. My understanding is that y_i is the *number* of individuals in population i who were infected with the VoC?

Line 94: what is k? On line 178, A_k is used to denote an area. Is this the same variable k?

Line 117: “if we consider them connected in some sense” -> “if one considers”

Line 278: please provide a reference to support the claim that London, Manchester, and Birmingham were among the first to be infected by Omicron

Line 405: inconsistent use of present and past-tense: “is found” vs “were small”

Line 411: “Considerable geographical heterogeneity was found for the probability that a test is sequenced or provided gene targets” -> I think this sentence is missing a word somewhere

Line 426: unexpected comma

**Have the authors made all data and (if applicable) computational code underlying the findings in their manuscript fully available?**

Reviewer #1: Yes

Reviewer #2: None

PLOS authors have the option to publish the peer review history of their article (what does this mean?). If published, this will include your full peer review and any attached files.

Reviewer #1: No

Reviewer #2: No

Figure Files:

Data Requirements:

Reproducibility:

References:

---

## [Decision Letter · Decision Letter 1]

9 Oct 2023

Dear Mr Ward,

We are pleased to inform you that your manuscript 'Bayesian Spatial Modelling of Localised SARS-CoV-2 Transmission Through Mobility Networks Across England' has been provisionally accepted for publication in PLOS Computational Biology.

Best regards,

Joseph T. Wu

Academic Editor

PLOS Computational Biology

Virginia Pitzer

Section Editor

PLOS Computational Biology

Reviewer's Responses to Questions

**Comments to the Authors:**

Reviewer #2: The reviewers have addressed my issues. I look forward to seeing this paper in print

**Have the authors made all data and (if applicable) computational code underlying the findings in their manuscript fully available?**

Reviewer #2: Yes

PLOS authors have the option to publish the peer review history of their article (what does this mean?). If published, this will include your full peer review and any attached files.

Reviewer #2: **Yes: **Jordan Douglas

---

## [Editor Report · Acceptance letter]

3 Nov 2023

PCOMPBIOL-D-23-00169R1 

Bayesian Spatial Modelling of Localised SARS-CoV-2 Transmission Through Mobility Networks Across England

Dear Dr Ward,

I am pleased to inform you that your manuscript has been formally accepted for publication in PLOS Computational Biology. Your manuscript is now with our production department and you will be notified of the publication date in due course.

With kind regards,

Livia Horvath
